# Monitoring and Analysis of the Collapse Process in Blasting Demolition of Tall Reinforced Concrete Chimneys

**DOI:** 10.3390/s23136240

**Published:** 2023-07-07

**Authors:** Xiaowu Huang, Xianqi Xie, Jinshan Sun, Dongwang Zhong, Yingkang Yao, Shengwu Tu

**Affiliations:** 1College of Science, Wuhan University of Science and Technology, Wuhan 430065, China; 2State Key Laboratory of Precision Blasting, Jianghan University, Wuhan 430056, China

**Keywords:** reinforced concrete chimney, blasting demolition, test monitoring, dynamic response, security control

## Abstract

Aiming at the problem of displacement of collapse direction caused by the impact of the high-rise reinforced concrete chimney in the process of blasting demolition, combined with the monitoring methods such as high-speed photography observation, piezoelectric ceramic sensor, and blasting vibration monitor, the impact process of the 180 m high chimney was comprehensively analyzed. The results show that the chimney will experience multiple ‘weight loss’ and ‘overweight’ effects during the sit-down process, inducing compressive stress waves in the chimney. When the sit-down displacement is large, the broken reinforced concrete at the bottom can play a significant buffering effect, and the ‘overweight’ effect gradually weakens until the sit-down stops. The stress of the inner and outer sides of the chimney wall is obviously different in the process of collapsing and touching the ground. The waveform of the monitoring point of the piezoelectric ceramic sensor is divided into three stages, which specifically characterizes the evolution process of the explosion load and the impact of the chimney. The vibration induced by explosive explosion is mainly high-frequency vibration above 50 Hz, the vibration induced by chimney collapse is mainly low-frequency vibration below 10 Hz, and the vibration characteristics are obviously different. In the process of blasting demolition and collapse of high-rise reinforced concrete chimney, due to the impact of sitting down, the wall of the support tube is subjected to uneven force, resulting in the deviation of the collapse direction. In practical engineering, the control measures of chimney impact, blasting vibration, and collapse touchdown vibration should be fully strengthened to ensure the safety of the protection target around the blasting demolition object.

## 1. Introduction

At present, the abandoned reinforced concrete chimneys are mainly demolished by blasting technology. In recent years, during the demolition of 180 m tall chimneys, some chimneys will have serious sitting problems; the middle of the chimney will be broken soon after detonation; the upper and lower sections will sometimes be separated; the collapse direction of the upper section of the chimney may be out of control; and—due to the change of the center of gravity position of the lower section of the chimney—it is easy to cause the lower part of the chimney to collapse in the opposite direction or “blow down”. This phenomenon may lead to serious security accidents.

The demolition process of chimney blasting involves many aspects, such as blasting notch formation, support zone failure, stress redistribution, and structural dynamic response, and its mechanical mechanism is complicated. In view of the failure mechanism of the support parts after the formation of the chimney blasting incision, Chu Huaibao et al. [1] observed the stress state of the retaining cylinder wall after the formation of the blasting incision, and believed that there was a process of load redistribution and neutral axis formation of 0.5~3.0 s after the formation of the blasting incision. Zheng Bingxu et al. [2] observed the support zone of six reinforced concrete chimneys cut and concluded that the support zone was prone to fall down due to the sudden load of self-weight after blasting. The incision end is first damaged by compression, and then the neutral axis is strained, showing a brittle fracture characteristic of large eccentric compression. Zheng Bingxu et al. [3] also analyzed the compression range of the support area caused by the sudden loading of the wound weight, and concluded that the central Angle of the incision should be 210~230°. Xu Pengfei et al. [4] believed that the neutral axis stability time of 2 to 3 s after the formation of blasting incision is the key to prevent premature sitting of the chimney and the formation of directional tipping trend. Yan Zhixin et al. [5] established a stress calculation model for the support zone after blasting of a reinforced concrete chimney and proposed the introduction of a stamping coefficient to consider the influence of sudden loading. Zhou Jiahan et al. [6] revised the ground vibration prediction formula based on the understanding of the ground touch process of blasting demolition of tall chimneys. Wang Yu et al. [7] studied the blasting demolition of 180 m reinforced concrete chimneys with weak drum walls; they concluded that the directional window Angle would first be damaged by compression, and an ideal “steering axis” would not be formed when the support area was insufficient, resulting in compression failure. According to the vibration monitoring results, Pu Zhiyou et al. [8] concluded that the vibration propagation velocity in the tipping direction is greater than the lateral vibration propagation velocity when the chimney is dismantled and hits the ground under the same geological conditions. Shuren Wang et al. [9] calculated and analyzed the measured data of chimney demolition and concluded that the attenuation law of vibration velocity presents a negative exponential power function from near to far.

Regarding the dynamic mechanical behavior monitoring of reinforced concrete materials under impact load, Zhong Dongwang [10] et al. applied piezoelectric ceramic sensors in the demolition and blasting of water towers to monitor and record the damage and damage degree at each stage and position. Si Jianfeng et al. [11] adopted the piezoelectric ceramic sensor active monitoring method to collect the active signal of the key point of the concrete sample along the central axis of the gun hole, indicating that the piezoelectric signal damage monitoring method can be used to monitor the damage depth of rock mass in underwater drilling and blasting. Huo [12] introduced piezoelectric-based bond-slip monitoring methods including the active sensing method, the electro-mechanical impedance (EMI) method, and the passive sensing using acoustic emission (AE) method, and the fiber-optic-based bond-slip detecting approaches—including the fiber Bragg grating (FBG) and the distributed fiber optic sensing—are highlighted, which provides guidance for practical applications and future development of bond-slip monitoring. Karayannis [13] presented the use of piezoelectric lead zirconate titanate (PZT) transducers for the examination of the efficiency of an innovative strengthening technique of reinforced concrete (RC) columns and BCJs. Shao-Fei Jiang [14] developed a monitoring approach by using piezoelectric-based smart aggregates (SAs) and an evaluation method with the damage extent (Dc) of crack was proposed based on the energy attenuation of stress wave.

Aiming at the dynamic response of the chimney structure under the action of longitudinal waves, Wang Yunjian [15] experimentally studied the relationship between the fracture position of the chimney, the shock wave action period, and the natural period of the chimney. Pallares et al. [16] used a three-dimensional finite element model to analyze the failure phenomenon of a masonry chimney under the action of earthquake, and obtained the failure mode, maximum stress, and displacement characteristics of the structure. Wolf et al. [17] studied the response characteristics of typical chimneys of nuclear power plants under earthquake and shock loads. Wilson [18] proposed a nonlinear dynamic analysis method for reinforced concrete chimneys based on the nonlinear characteristics of 10 chimneys under earthquake action. Huang et al. [19] proposed a new 3D nappe analysis method based on the seismic dynamic response of reinforced concrete chimneys with a height of 115 m. Minghini et al. [20] analyzed the failure of brick chimneys during earthquakes and expounded the shear failure mechanism of the upper part of chimneys.

To sum up, there have been numerous studies on the instability, collapse, motion process, and seismic response of reinforced concrete chimneys demolished by blasting, but there are only a few studies on the down-sitting of chimneys during blasting demolition. At the same time, the relevant technology of the piezoelectric ceramic sensor for monitoring the dynamic mechanical response of reinforced concrete has become very mature, which can sensitively capture the dynamic characteristics of the material. In this paper, through the observation and analysis of the movement and fracture process of a 180 m high reinforced concrete chimney, combined with the monitoring results of the embedded piezoelectric sensor, the phenomenon of the chimney demolition is analyzed and discussed.

## 2. Engineering Cases

The reinforced concrete chimney demolished by blasting is 180 m high and the concrete label is C30. The main structural dimensions of the chimney are shown in Table 1. The transverse axial reinforcement of the chimney is double reinforcement, with the outer vertical reinforcement Φ22@200, and the inner vertical reinforcement Φ16@200; the ring is equipped with horizontal stirrup, the outer ring rib Φ18@200, and the inner ring rib Φ14@200. There are two smoke vents at the bottom of the chimney. The center line of flue 1 is located 60° to the west of the north at the bottom of the chimney, with elevation of +0.46 m~+5.78 m, height of 5.32 m, and width of 5.40 m. The center line of flue 2 is located in the north direction of the bottom of the chimney, with elevation of +7.50 m~+12.82 m, height of 5.32 m, and width of 5.40 m.

The chimney adopts a positive trapezoidal blasting incision, as shown in Figure 1. The incision is 6.0 m high and is arranged at the bottom of the chimney at an elevation of 0.5 m. The central Angle of the blasting incision is 216°, and the base edge is 34.9 m long. Directional windows are arranged on both sides of the incision, with a base length of 2.0 m and an opening of 30°. A total of 612 holes are arranged in the blasting incision area, the total charge of emulsion explosive is 124 kg; the prompt detonating tube detonator is arranged in the gun hole and all charge packs explode at one time.

## 3. Monitoring System and Sensors

### 3.1. High-Speed Photographic Observation

In order to analyze the instability and failure process of the chimney, a set of dynamic photogrammetry system is arranged outside the support area of the chimney. The photogrammetric system is composed of a high-precision industrial camera, benchmark ruler, measuring mark, calculation software, and computer. The xyz coordinate value of the target point is collected by the measuring camera first, and then the standard fitting of the point line and surface is undertaken by the target point. Two miniature high-speed cameras with a resolution of 1280 × 960 and a acquisition frequency of 100 s^−1^ were symmetrically installed on the supports on the ground at the back of the support area (see Figure 1). At the same time, a high-speed camera is arranged directly behind the chimney barrel and the acquisition frequency is set to 5000 s^−1^. In the remote area of blasting, the UAV is used to record the blasting process in the air.

### 3.2. High-Speed Photographic Observation

Piezoelectric ceramic materials are widely used in structural health monitoring and damage detection. They have the characteristics of low cost, wide frequency response, and can be used as both transmitting and receiving sources. Piezoelectric ceramics with small capacitance and high resistance value are commonly used to process sensors, which have been widely welcomed in daily life and scientific research. In order to monitor the dynamic response characteristics of the chimney wall in the process of blasting demolition and collapse, three monitoring holes are arranged symmetrically along the designed collapse center of the chimney in the chimney support area, and two piezoelectric ceramic sensors are arranged in each monitoring hole along the inner and outer sides, respectively, totaling six sensors. The piezoelectric ceramic sensor is distributed, as shown in Figure 2. The three monitoring holes were marked successively from right to left as 1# measuring points (CH1 and CH2), 2# measuring points (CH3 and CH4), and 3# measuring points (CH5 and CH6); the measuring points on the inside of the cylinder wall were CH1, CH3, and CH5; the measuring points on the outside of the cylinder wall were CH2, CH4, and CH6; and the sampling rate was set at 1 KHZ. Before the piezoelectric ceramic sensor is placed into the monitoring hole, it is fixed on the plaster strip with 502 glue to maintain the design distance, and the sensor is buried, as shown in Figure 3.

### 3.3. Blast Vibration Monitoring

Select representative buildings close to the explosion area for vibration monitoring. The vibration monitoring equipment mainly adopts the Micromate vibrometer of the Canadian INSTANTEL brand. The main technical indicators are: (1) vibration range: 254 mm/s; (2) vibration resolution 0.00788 mm/s; (3) vibration linear accuracy: +/−0.5 mm/s; (4) frequency range 2–250 Hz; and (5) sampling frequency: program-controlled switching of 1024, 2048, 4096 Hz per channel. The vibration sensor and the cement floor are coupled with gypsum powder, as shown in Figure 4. According to the surrounding environment, a total of three vibration measurement points are arranged, among which the measurement point 1 is set on the civil building, 24 m away from the chimney; measuring point 2 is set in the pipe support of the gas pipeline, 35 m away from the chimney; and measuring point 3 is also set on the pipe support, 145 m away from the chimney. Due to the complex surrounding environment of the chimney, the monitoring line layout conditions of attenuation law are not available, so the monitoring is mainly protective monitoring, and the location diagram of the measuring point is shown in Figure 5.

### 3.4. Damage Monitoring Based on Piezoelectric Ceramics

Piezoelectric ceramics have positive and inverse piezoelectric effects. The piezoelectric ceramic transducer is implanted inside the structure or pasted on the surface of the structure, and the structure under test can together form a health monitoring system. The piezoelectric ceramic drive unit is excited by a selected signal and, due to the inverse piezoelectric effect, the piezoelectric ceramic vibrates and stress waves propagate on the surface or inside the structure under test, and the piezoelectric ceramic transducer at a certain distance apart, due to the positive piezoelectric effect, converts the measured stress waves into electrical signals. Due to the effect of damage in the structure, the signal amplitude, energy, propagation time, mode, and waveform will be changed. Comparing the signal in the healthy state of the structure can determine whether the structure is damaged, and further analysis can also identify the location and extent of damage. Based on the brittle characteristics of PZT material, PZT is usually encapsulated in engineering tests to meet different working conditions. As shown in Figure 6, the PZT is encapsulated in a metal housing with a diameter of φ2.5 cm and a height of 1 cm in this study; an amplification circuit is added to the sensor considering the amplification of the signal, and the amplification circuit is separated from the PZT by a magnetic steel bedding layer. The sensor is connected to a shielded cable, which is connected to the collector or signal generating instrument through BNC connectors. Figure 7 shows the schematic diagram of the structure of the health monitoring system based on PZT active sensing.

## 4. Analysis of Chimney Collapse Impact Monitoring Results

### 4.1. Image Analysis of Cylinder Wall Down-Sit Impact Process

From the monitoring video of the blasting process, it can be seen that the concrete on both sides of the incision after detonation was strongly squeezed and destroyed. At about 0.5 s after detonation, cracks with an Angle of 45° from the horizontal direction were generated in the support area from the Angle of the directional window (Figure 8a) and expanded from both sides to the middle. With the expansion of the main crack, a large amount of concrete is constantly extruded from the wall and falls off. About 1.2 s after detonation, cracks spread along the cylinder wall in a circular direction (Figure 8b). About 2.0 s after detonation, the main fracture in the support zone was connected (Figure 8c). About 4.5 s after initiation, the blasting incision is completely closed (Figure 8d).

The monitoring results show that the cracks spread faster in the back of the support area and the concrete damage is more serious. During the process of fracture generation and expansion to penetration, no obvious tensile fracture was observed at the back of the support area, indicating that the failure mode of the support area of the chimney was mainly compressive shear failure. After the formation of cracks through the chimney, the support area cannot bear the load of the superstructure and the whole sit down. In the process of sitting down, the chimney barrel constantly collided with the bottom residual structure and the ground, and the concrete at the bottom of the barrel was crushed and extruded and piled up around the chimney. About 4.5 s after initiation, the chimney down process is over. During the whole sitting process, the sitting speed of the chimney first increased and then quickly slowed down to 0, which lasted about 2.5 s, and the total height of the sitting was 8.3 m. At the same time, the chimney also produced a slight rotation during the process of sitting down, but the overall rotation Angle was not large, and the chimney rotation Angle was about 3° at the end of the lower seat.

Through photogrammetry, the displacement time history curve of the chimney down-sitting process was obtained, as shown in Figure 9. The measured data show that, after the blasting of the chimney, the amount of sitting down increases slowly with time, then increases rapidly, and finally stops gradually. After first and second differentiation of the downward displacement time history curve to time t, the downward velocity (Figure 10) and acceleration time history curve (Figure 11) can be approximated. From 1.0 to 1.5 s after the start of sitting down, the sitting speed can reach 5.96 m/s and the acceleration of the downward movement can be close to 6.4 m/s^2^. When the downward motion is decelerated, the maximum acceleration of the upward motion can be close to 8.8 m/s^2^. As a result, the chimney may experience multiple “weightlessness” and “overweight” effects during sitting down. The “weightlessness” and “overweight” change faster, will induce compressive stress waves in the chimney, and may cause the chimney to continue to sit down. When the sitting volume is large, the broken reinforced concrete at the bottom can play a significant buffer effect, and the “overweight” effect gradually weakens until the sitting down stops.

### 4.2. Analysis of the Dynamic Response Characteristics of the Cylinder Wall

The monitoring data in the frequency range of 0~1 kHz are selected to read the voltage amplitude results collected by six piezoelectric ceramic sensors in three monitoring points on the chimney wall, as shown in Table 2.

It can be seen from the data in the table that there are obvious differences in the values of the inner and outer walls of the chimney, indicating that the forces of the inner and outer walls of the chimney are obviously different during the process of the chimney collapsing and touching the ground. Among them, the extreme value of voltage monitoring at 1# and 2# was between 1079 and 1979 mV, respectively, which was significantly higher than that at 3# (130.28 to 208.80 mV). The results showed that the chimney did not fall in strict accordance with the designed collapse center line during the collapse process, resulting in the support zone cylinder wall not being uniformly stressed but biased towards the 1# measurement point. The monitoring results were in good agreement with the image observation results.

Read the waveforms of the monitoring points of three piezoelectric ceramic sensors, as shown in Figure 12. It can be seen from the waveform that the monitoring waveform is obviously divided into three stages in the time domain, and the time domain performance of the voltage waveform is in good agreement with the image observation results. Among them, the first phase of the waveform is located between 0.5 and 0.7 s after the chimney blasting. At the moment when the chimney blasting incision is formed, the piezoelectric ceramic sensor senses the stress change of the cylinder wall and the dynamic response is mainly stimulated by the explosion load. The 2# measuring point is located at the center line of the chimney collapse, and the voltage monitoring amplitude is abnormally prominent compared with the monitoring points on both sides, mainly due to the superposition of the explosion stress waves on both sides. The second phase of the waveform is located between 0.7 and 3.0 s after the initiation of the chimney, and the chimney support cylinder wall begins to crack and penetrates from both sides to the middle. At this stage, the voltage waveform of the piezoelectric ceramic sensor tends to be flat, indicating that there is no obvious force. The third phase of the waveform is located between 3.0 and 4.5 s after the initiation of the chimney, and the chimney support cylinder wall begins to sit down after the crack is penetrated and the piezoelectric ceramic sensor is impacted, so the voltage signal is very significant. Among them, the voltage amplitude of the 1# measuring point is larger and shorter in the time domain, followed by the 2# measuring point, and the 3# measuring point has the smallest amplitude and is longer in the time domain. This shows that the impact process of the lower sitting of the chimney is an evolutionary process from sharp to slow.

### 4.3. Blasting Vibration Characteristics Analysis

The damage caused by blasting vibration to buildings mainly means that the vibration wave generated by blasting propagates to the base of the building through the ground, and then transmits to the superstructure of the building through the base, causing the vibration of the building. The measured vibration data show that, in the process of chimney blasting demolition, the total duration of vibration is 18.5 s, the vibration induced by explosive explosion is mainly high-frequency vibration above 50 Hz, and the vibration induced by chimney barrel collapse and contact with the ground is mainly low-frequency vibration below 10 Hz. The time history curve is shown in Figure 13. The peak vibration velocity of blasting vibration and the peak vibration velocity of touchdown vibration are both the largest at the measuring point 1 closest to the blasting location, and the peak vibration velocity of blasting vibration particle decreases gradually with the increase of distance and the continuation of time.

In order to simplify the model, a representative measuring point 1 is selected to draw the maximum response spectrum of a system of elemental points with the same damping and different periods, and the relationship between structural dynamics and blasting vibration characteristics is comprehensively considered.

The equation of motion of a single degree of freedom system under uniform ground excitation is:(1)$$\ddot{y}+2\zeta \omega \dot{y}+\omega^{2}y={\ddot{x}}_{g}(t)$$
where, *ζ*—damping; *ω*—intrinsic frequency; $y,\;\dot{y},\ddot{y}$—relative displacement, velocity, and acceleration of mass points, respectively; and ${\ddot{x}}_{g}(t)$—seismic acceleration.

The steady-state solution of *y* can be expressed in the following Duhamel integral form:(2)$$y(t)=-\frac{1}{w_{d}}\int_{0}^{t}e^{-\xi \omega (t-\tau )}{\ddot{x}}_{g}(\tau )\sin w_{}(t-\tau )d\tau$$
where there is a damped circle frequency $w_{d}=w\sqrt{1-\zeta^{2}}$. To solve the primary and secondary differentiation of Equation (2)—that is, to obtain the standard velocity and acceleration response equations for the single degree of freedom system—in general, the damping ratio is taken to be within 0.2. Ignoring the damping ratio *ζ* product term, the differentiation is simplified as
(3)$$\dot{y}(t)=\left[\frac{w}{w_{d}}\int_{0}^{t}e^{-\xi \omega (t-\tau )}{\ddot{x}}_{g}(\tau )\cos\omega (t-\tau )d\tau \right]/{\dot{x}}_{g}(t)$$
(4)$$\ddot{y}(t)=\left[\frac{w^{2}}{w_{d}}\int_{0}^{t}e^{-\xi \omega (t-\tau )}{\ddot{x}}_{g}(\tau )\sin\omega (t-\tau )d\tau \right]/{\ddot{x}}_{g}(t)$$

Data processing software was used to solve the response spectrum of measuring point 1 with a large vibration amplitude, and the response spectrum curves of displacement, velocity, and acceleration were drawn, as shown in Figure 14.

By comparing each response spectrum curve, it can be seen that: (1) With the change of damping ratio of buildings, the response to blasting vibration will also change. With the extension of vibration action time, the displacement response spectrum first rises to the peak, and then falls to a stable range and then oscillates. The velocity response spectrum rises steadily to the maximum value, then gradually decreases and finally tends to equilibrium. The acceleration response spectrum reaches a large peak value at the initial stage and then gradually decays to zero. (2) Within 0.4 s of blasting action, it is the reactive activity area of the building, and then gradually attenuates until it becomes stable, indicating that the impact of blasting vibration on the building is mainly in a certain time after the explosive explosion. (3) The damping coefficient has a great influence on the response spectrum. With the increase of the damping ratio, the peak value of the vibration response gradually decreases, indicating that the damping ratio has the effect of eliminating the peak. (4) The peak tip frequency band of the displacement and velocity response spectrum is wider than that of the acceleration peak tip frequency band, and the displacement response spectrum has the largest fluctuation.

## 5. Conclusions

In this paper, a comprehensive analysis of the downward impact process of a 180 m tall reinforced concrete chimney is carried out by means of high-speed photographic observation, piezoelectric ceramic sensor, and blasting vibration monitor. The following conclusions and recommendations are drawn:Tall chimneys will experience multiple “weightlessness” and “overweight” effects after blasting, inducing compressive stress waves in the chimneys. When the sitting displacement is large, the broken reinforced concrete at the bottom can play a significant cushioning role, and the “overweight” effect gradually weakens until the sitting down stops.The piezoelectric ceramic sensor embedded in the chimney can sensitically capture the dynamic stress information of the chimney wall, and the monitoring data show that the forces inside and outside the chimney wall have obvious differences during the collapse and touching the ground. The waveforms at the monitoring points of the piezoelectric ceramic sensor are generally divided into three stages, which specifically represent the evolution process of the explosion load and the downward sitting impact of the chimney.The vibration induced by explosive explosion is mainly high-frequency vibration above 50 Hz, the vibration induced by chimney barrel collapse and touching the ground is mainly low-frequency vibration below 10 HZ, and the vibration characteristics have obvious differences. The response spectra of displacement, velocity, and acceleration corresponding to different damping ratios are obtained by analyzing the response spectra of blasting vibration signals. The difference of the three maps shows that the buildings with different damping coefficients have different responses to blasting vibration.In the process of demolition and collapse of tall reinforced concrete chimneys, due to the impact of sitting down, the supporting cylinder wall is stressed unevenly, causing the collapse direction to shift. In practical engineering, the control measures of the down-sitting impact, blasting vibration, and collapse vibration of the chimney should be fully strengthened to ensure the safety of the protection target around the blasting demolition object.

## Figures and Tables

**Figure 1 sensors-23-06240-f001:**
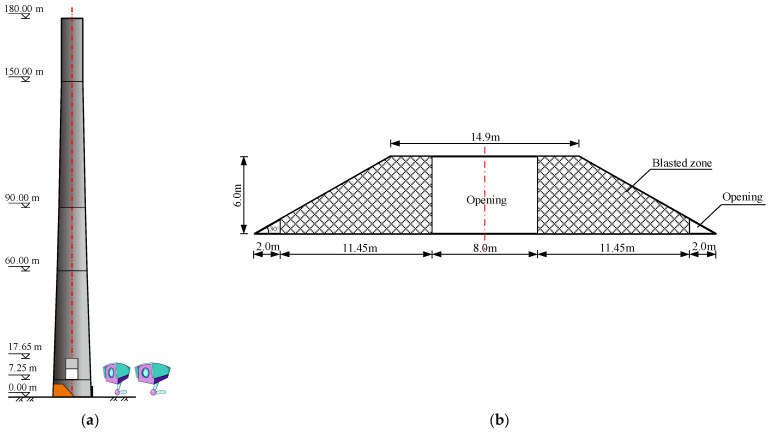
Blasting plan of the chimney; (**a**) Arrangement of the blasting notch; (**b**) Parameters of the blasting notch.

**Figure 2 sensors-23-06240-f002:**
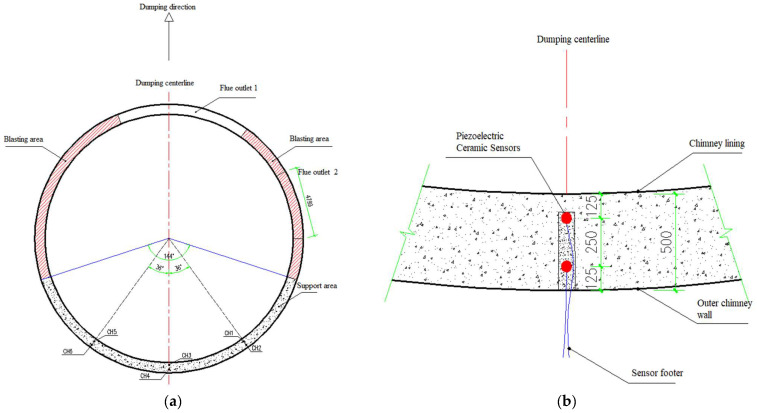
Schematic diagram of piezoelectric ceramic sensor arrangement; (**a**) Sensor distribution; (**b**) Sensor distribution position (unit: mm).

**Figure 3 sensors-23-06240-f003:**
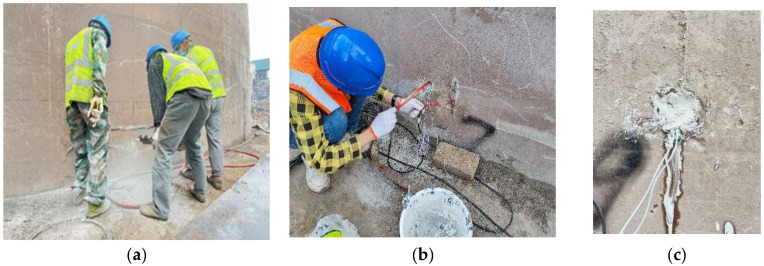
Chimney cylinder wall monitoring point arrangement process; (**a**) Drilling of monitoring holes; (**b**) Pre-embedded sensors; (**c**) Plugging and grouting.

**Figure 4 sensors-23-06240-f004:**
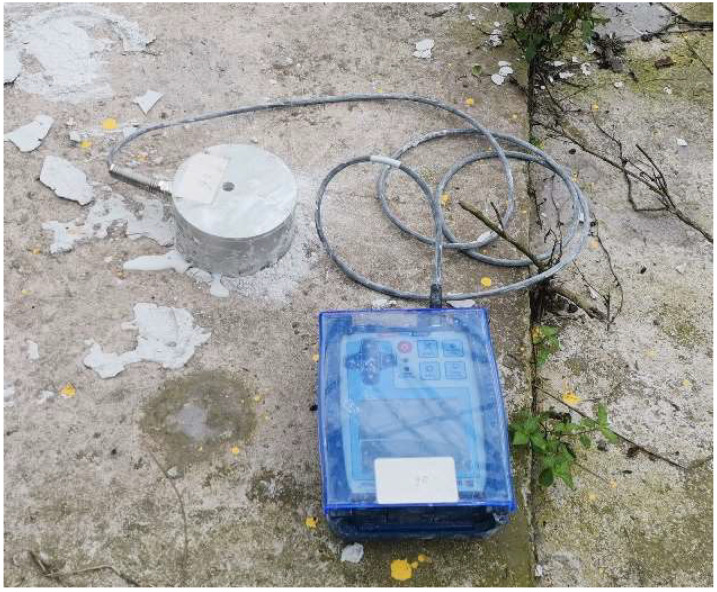
Micromate type vibration monitor.

**Figure 5 sensors-23-06240-f005:**
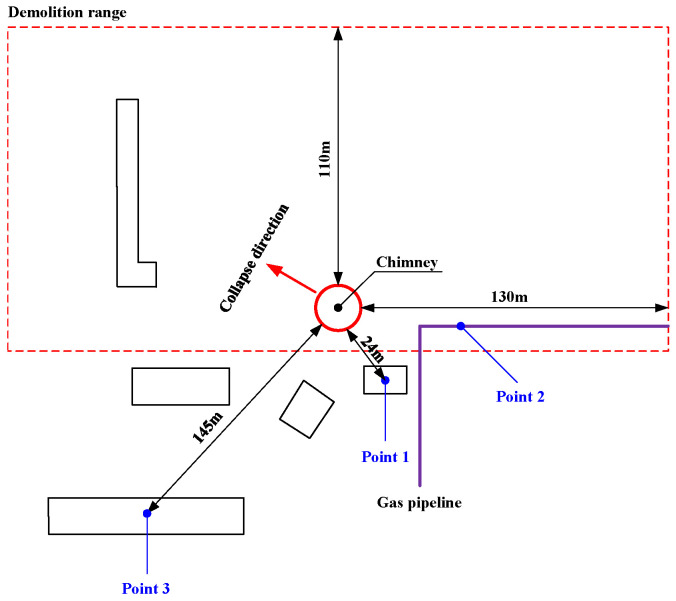
Blasting vibration measurement point layout diagram.

**Figure 6 sensors-23-06240-f006:**
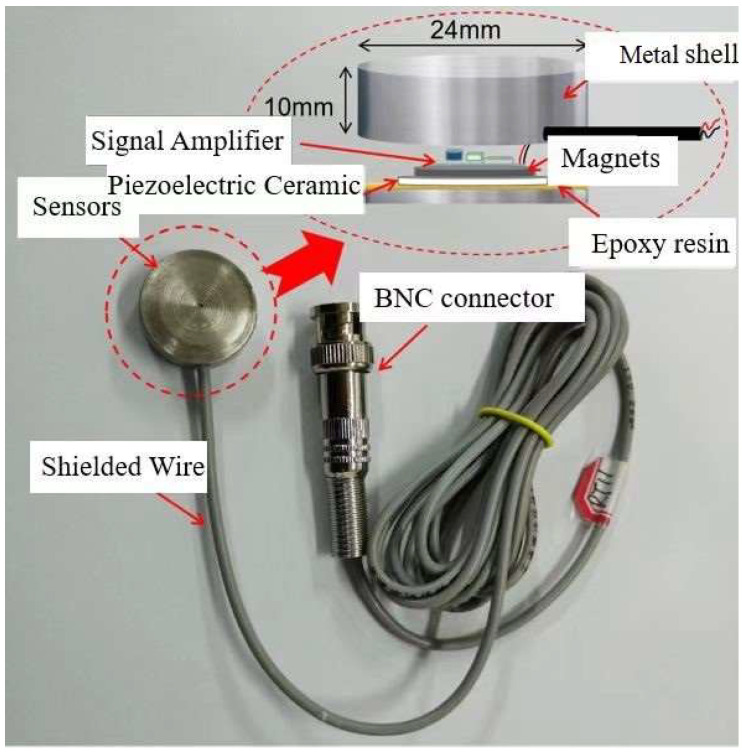
Basic structure of piezoelectric ceramic sensors.

**Figure 7 sensors-23-06240-f007:**
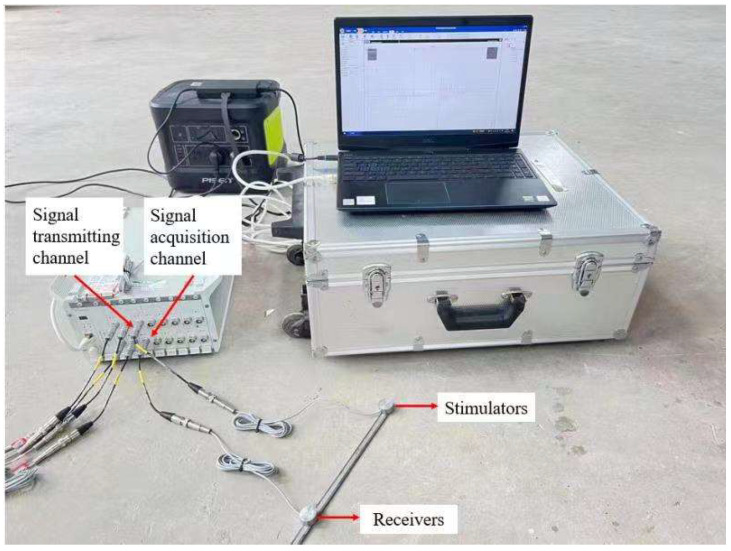
Piezoelectric signal monitoring and analysis system.

**Figure 8 sensors-23-06240-f008:**
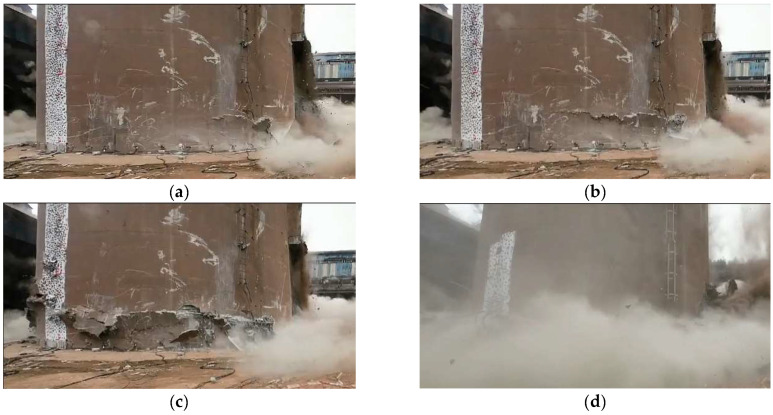
Crack propagation process in the support part; (**a**) 0.5 s; (**b**) 1.2 s; (**c**) 2.0 s; (**d**) 4.5 s.

**Figure 9 sensors-23-06240-f009:**
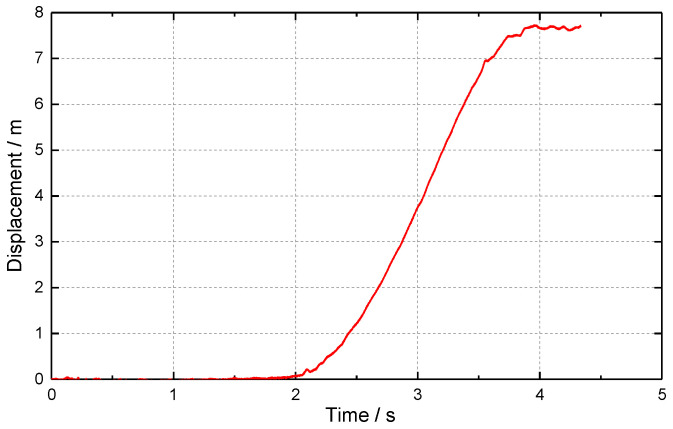
Displacement-time history curve of chimney sitting down.

**Figure 10 sensors-23-06240-f010:**
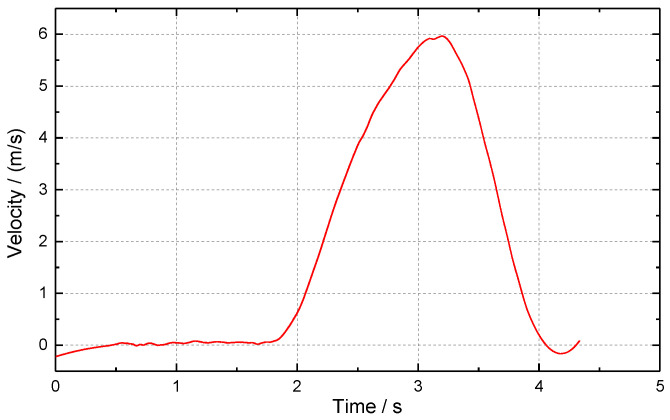
Velocity-time history curve of chimney sitting down.

**Figure 11 sensors-23-06240-f011:**
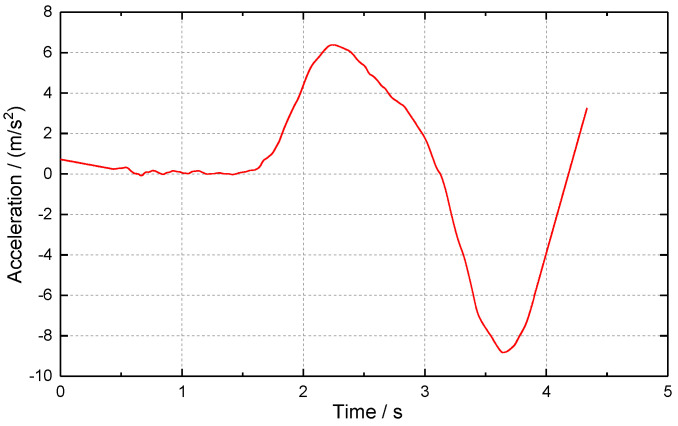
Acceleration-time history curve of chimney sitting down.

**Figure 12 sensors-23-06240-f012:**
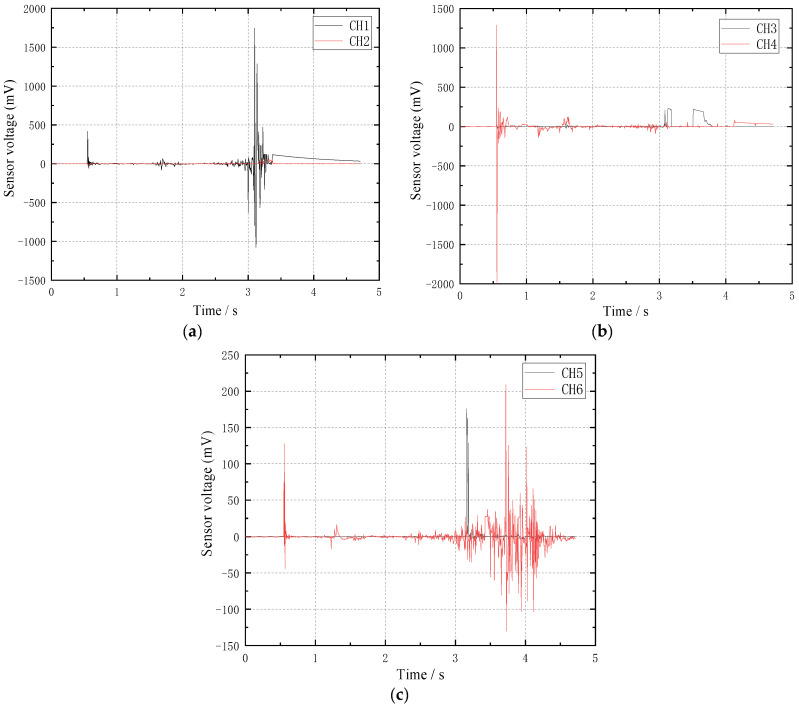
Piezoelectric ceramic monitoring point waveform diagram; (**a**) 1# measuring point; (**b**) 2# measuring point; (**c**) 3# measuring point.

**Figure 13 sensors-23-06240-f013:**
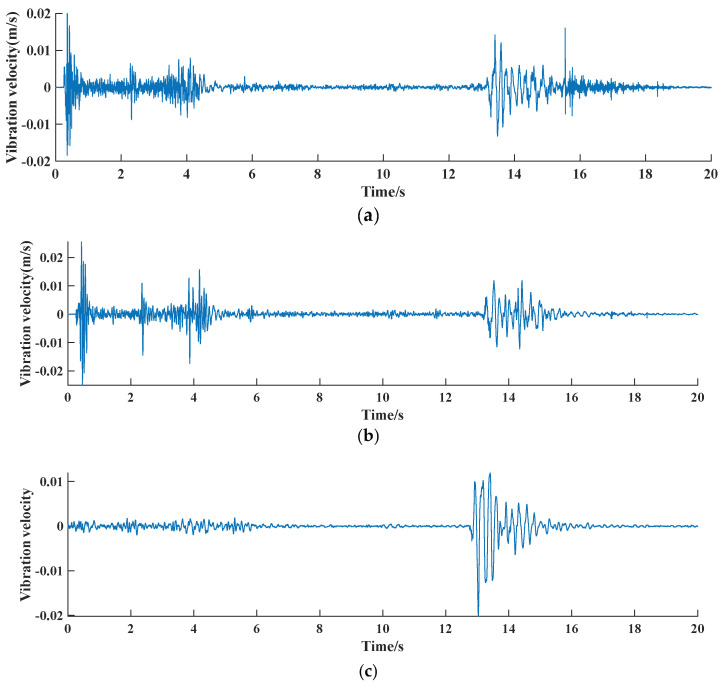
Time course curve of vibration velocity at each measurement point; (**a**) 1# measuring point; (**b**) 2# measuring point; (**c**) 3# measuring point.

**Figure 14 sensors-23-06240-f014:**
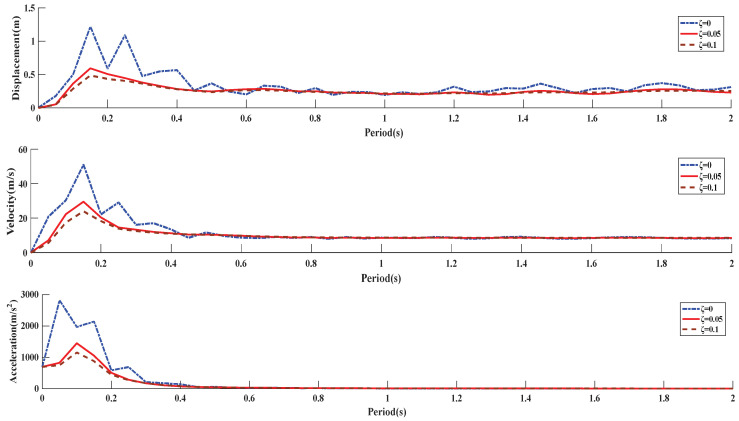
Response spectrum of parameter variation with damping.

**Table 1 sensors-23-06240-t001:** Structure parameters of the chimney.

Elevation/m	Outer Radius of Cylinder Wall/cm	Inner Radius of Cylinder Wall/cm	Thickness of Cylinder Wall/cm	Thickness of Insulation Layer/cm	Thickness of Lining/cm
0.00	925	875	50	8	24
30.00	745	697	48	8	24
60.00	595	553	42	8	12
90.00	505	469	36	8	12
120.00	425	395	30	8	12
150.00	365	341	24	8	12
180.00	305	285	20	8	12

**Table 2 sensors-23-06240-t002:** Piezoelectric ceramic sensors collect data amplitude.

Measurement Point Number	Sensor Number	Sensor Location	Maximum Value		Minimal Value	
			Magnitude/mV	Time/s	Magnitude/mV	Time/s
1#	CH1	Inner	1745.71	3.10	−1079.40	3.12
	CH2	Outer	54.16	2.74	−21.55	0.56
2#	CH3	Inner	230.33	3.14	−24.50	1.60
	CH4	Outer	1288.08	0.54	−1979.92	0.55
3#	CH5	Inner	175.93	3.16	−3.28	3.95
	CH6	Outer	208.80	3.72	−130.28	3.73

## Data Availability

Not applicable.

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
