# Peer review of "Monitoring and Analysis of the Collapse Process in Blasting Demolition of Tall Reinforced Concrete Chimneys"

_sensors, 2023, doi:10.3390/s23136240_

Round 1

Reviewer 1 Report

The manuscript displays a case study of the impact process of a reinforced concrete chimney barrel. The topic is interesting, but the analysis, figure quality, equation format, and especially the English expression must be further improved before consideration.  

1. It is more like a report rather than a journal article, authors must pay more effort to the analysis of the results from different views. 2. The Abstract is too long with some unimportant results. 3. The quality of figures and equations should be improved. 4. What is the novelty of this manuscript, should mention in the abstract and introduction. 5. The reference format should be consistent.

The English expression must be improved.

Reviewer 2 Report

The submitted Essay with the Manuscript ID: sensors-2448417 and the Title: "Monitoring and Analysis of Collapse Process in Blasting Demolition of Tall Reinforced Concrete Chimneys" is an interesting contribution to the implementation of a combination of several monitoring methods such as high-speed photography observation, piezoelectric ceramic sensor and blasting vibration monitor to analyze the impact process, the phenomenon of down-sitting and early aerial fracture of a reinforced concrete (RC) chimney barrel blasting demolition. The paper falls within the scope of the Journal and deals with a topic still open to question. The work is comprehensive, and the paper is well-organized and well-written. The following comments and suggestions are raised for the authors' reference:

1. The literature review is extensive and informative in the introduction. However, the following relevant topics are suggested to be considered since they might contribute to the promotion of the aims of this study:

(a) The proposed structural health monitoring (SHM) method, which is based on piezoelectric ceramics and presented in sub-section 3.4, uses externally installed piezoelectric ceramic sensors (PZT) encapsulated in a metal housing. The implementation of externally surface adhesively bonded PZTs in SHM techniques has recently been extended for the damage detection, identification, and assessment of their severity level in large-scale RC structural members under shear and flexural monotonic and cyclic loading. The ability of this SHM method to qualitatively identify damage caused by concrete cracking and steel yielding in flexural beams has also been established.

(b) The developed SHM method suggests the application of a single encapsulated PZT sensor. However, recent studies showed that using a network of small-sized PZTs, instead of individual (single) transducers provides more accurate SHM results. The advantages of this methodology should be emphasized. The proposed procedure to correct the sensor's sensitivity could be established further based on previous works.

(c) A recently developed small-sized, portable, and wireless SHM system has been successfully applied for efficient flexural damage diagnosis in RC beams using a wireless admittance monitoring system. Some aspects of this PZT-based SHM system could help to future improvement of the monitoring system described in the submitted study.

(d) Research significance and especially the subsequent impact of the presented study on the state of the practice is not adequately established. A systematic literature review based on the previous recommendations would help.

The following relative articles could be used as examples of relative articles that could support the issues above:

- "Bond-slip monitoring of concrete structures using smart sensors - A review", Sensors 2019.

- "Damage monitoring of concrete laminated interface using piezoelectric-based smart aggregate", Engineering Structures, 2021.

- "Efficacy and damage diagnosis of reinforced concrete columns and joints strengthened with FRP ropes using piezoelectric transducers", Sensors, 2022.

2. The experimental validation of the examined monitoring methods to analyze the impact process of the RC chimney is weak; thus, it is strongly recommended to enrich this part.

3. Some variables are not well-presented in several equations and should be corrected. A notation list should be added, too.

4. The frequency range in which the response signals of PZTs used for data acquisition seems to depend on a frequency-domain signal and occasionally cannot accurately reflect the load condition of the RC members. Thus, some clarifications and perhaps more comments to justify the selection of the adopted frequency ranges would be helpful.

5. Concluding remarks and further discussion concerning the combination of the examined SHM systems could be helpful to be added. A supplementary table summarizing each method's characteristics, merits, and shortcomings is also suggested to be included.

Round 2

Reviewer 1 Report

The authors tried to improve the quality of the manuscript, and most of the comments and suggestions are well addressed. I would like to recommend it for publication. 

Reviewer 2 Report

The questions and requirements posed by the review round 1 have been addressed acceptably. The revised manuscript with the Manuscript ID: sensors-2448417 and the Title: "Monitoring and Analysis of Collapse Process in Blasting Demolition of Tall Reinforced Concrete Chimneys" has extensively been improved with several clarifications and additional discussions as requested. The efforts performed by the Authors to consider all the recommendations and to respond to all the criticisms of the previous review comments are greatly appreciated. Hence, the paper is suggested to be accepted for publication in the journal without further re-review.